# Anti-Inflammatory and Anti-Thrombogenic Properties of Arterial Elastic Laminae

**DOI:** 10.3390/bioengineering10040424

**Published:** 2023-03-28

**Authors:** Jeremy Goldman, Shu Q. Liu, Brandon J. Tefft

**Affiliations:** 1Department of Biomedical Engineering, Michigan Technological University, Houghton, MI 49931, USA; 2Biomedical Engineering Department, Northwestern University, Evanston, IL 60208, USA; 3Department of Biomedical Engineering, Medical College of Wisconsin & Marquette University, Milwaukee, WI 53226, USA

**Keywords:** elastin, inflammation, thrombosis, intimal hyperplasia, arterial reconstruction

## Abstract

Elastic laminae, an elastin-based, layered extracellular matrix structure in the media of arteries, can inhibit leukocyte adhesion and vascular smooth muscle cell proliferation and migration, exhibiting anti-inflammatory and anti-thrombogenic properties. These properties prevent inflammatory and thrombogenic activities in the arterial media, constituting a mechanism for the maintenance of the structural integrity of the arterial wall in vascular disorders. The biological basis for these properties is the elastin-induced activation of inhibitory signaling pathways, involving the inhibitory cell receptor signal regulatory protein α (SIRPα) and Src homology 2 domain-containing protein tyrosine phosphatase 1 (SHP1). The activation of these molecules causes deactivation of cell adhesion- and proliferation-regulatory signaling mechanisms. Given such anti-inflammatory and anti-thrombogenic properties, elastic laminae and elastin-based materials have potential for use in vascular reconstruction.

## 1. Introduction

The wall of arteries consists of extracellular matrix components, including collagen matrix and elastic laminae. The essential functions of the extracellular matrix are to organize vascular endothelial cells, smooth muscle cells, and fibroblasts into the intima, media, and adventitia of the arterial wall, respectively; provide mechanical strength and elasticity to the arterial wall; and participate in cell signal transduction involved in vascular development and pathogenic processes such as inflammation, thrombosis, and atherosclerosis. Elastic laminae work with the collagen matrix in an antagonistic manner to control vascular cell and leukocyte adhesion, proliferation, and migration, which are cell activities directly influencing inflammatory, thrombogenic, and atherogenic processes [1,2,3,4]. Whereas the collagen matrix stimulates these cell activities, enhancing inflammatory, thrombogenic, and atherogenic processes [5,6,7], the elastic laminae exert an opposite effect, suppressing these pathogenic processes [8,9,10,11,12,13,14,15,16,17,18,19,20,21]. The antagonistic action of the elastic laminae helps prevent excessive inflammatory responses and vascular disorders [8,9,11,12,13,14,15,16,17,18,19,20,21]. In arterial reconstruction, these elastic lamina properties can prevent intimal hyperplasia, a process leading to restenosis and failure of arterial grafts. This paper reviews the role of the arterial elastic laminae in controlling inflammatory responses, thrombosis, and neointima formation in reconstructed arteries.

Inflammation in reconstructed arteries is a series of processes activated in response to surgery, mechanical injury, and exposure to biomaterials [1,22,23,24,25,26]. In the host artery near the junction with a reconstructed artery, several inflammatory processes can occur, including elevation in the endothelial permeability, interstitial edema, cytokine expression and secretion, leukocyte adhesion to injured endothelial cells, smooth muscle cell and fibroblast proliferation, extracellular matrix overproduction, and fibrosis. At the junction of the host artery and the reconstructed artery, blood coagulation may be induced in response to injury and hemorrhage. These processes contribute to thrombosis and intimal hyperplasia, resulting in the formation of neointima. In reconstructed arteries, inflammation and thrombosis occur during the early phase with the level dependent on the material at the blood-contacting surface [27,28,29]. For instance, in autologous vein grafts, the causes and pathological processes described above for the host artery occur. In synthetic material-based arterial grafts, blood coagulation can be induced rapidly, resulting in thrombosis (note that synthetic grafts can only be used under high flow conditions, which reduce the rate of thrombogenesis, and it is necessary to use anti-coagulants to minimize the risk of thrombus development). Following this phase, smooth muscle cells can proliferate and migrate from the host artery into the thrombus of the reconstructed artery, contributing to the development of neointima, which can cause restenosis of the reconstructed artery [30,31,32,33,34,35,36,37,38]. In autologous vein-based arterial constructs, smooth muscle cells in the venous wall can also proliferate and migrate into the thrombus to form neointima. Thus, a critical concern in arterial reconstruction is how to prevent inflammatory responses, thrombosis, and neointima formation. As the arterial elastic laminae exert an inhibitory effect on inflammatory responses, thrombosis, and smooth muscle cell proliferation and migration [8,9], this extracellular matrix and elastin-based materials can potentially be used as a surface material for arterial reconstruction.

## 2. Molecular Structure of Elastic Laminae

### 2.1. Elastin Gene

Elastin is a polymer, and its precursor, tropoelastin, is a protein encoded by the *ELN* gene in humans [39]. The *ELN* gene is a 45 kb segment within chromosome 7q11.1. It is comprised of 34 exons and nearly 700 introns [40]. Elastin consists of alternating hydrophobic and hydrophilic domains. The hydrophobic domains are rich in hydrophobic amino acids such as glycine and proline. These domains are important for the self-assembly of supramolecular structure. The hydrophilic domains are rich in lysine residues. These domains are important for crosslinking to form a highly stable, insoluble structure. 

Elastin production occurs primarily during fetal development and postnatal growth and is negligible by early adulthood [41,42,43]. *ELN* gene transcription is steady throughout the lifespan and elastin production is primarily regulated by posttranscriptional destabilization of tropoelastin mRNA in mature tissue [44,45]. The low synthesis and turnover of elastin in adults has important implications in aging and disease.

There are at least 11 isoforms of elastin as a result of alternative splicing of the *ELN* pre-mRNA [46,47]. These isoforms result in tissue-specific variants of elastin with unique properties [48]. The structure and function of these isoforms are subjects of ongoing research.

### 2.2. Tropoelastin

Tropoelastin is the soluble protein precursor to elastin and has a molecular weight of 60–70 kDa [49,50,51]. Once exported from the cell, tropoelastin molecules reversibly self-assemble into globular aggregates of elastin. Self-assembly is caused by interactions between the hydrophobic domains in a process known as coacervation [52,53,54]. This is followed by irreversible crosslinking of lysine residues within the hydrophilic domains. The crosslinking process involves the formation of desmosine and isodesmosine covalent crosslinks by the enzyme lysyl oxidase (LOX) [55]. Both inter- and intra-chain crosslinks are formed. Important for its mechanical characteristics, tropoelastin is sufficiently structured to self-assemble, yet sufficiently flexible to maintain elasticity. Elastin’s remarkable extensibility arises from the coil region near the N-terminus. This region acts like a spring, allowing tropoelastin to stretch up to eight times its resting length when free of crosslinks [56].

### 2.3. Elastic Fibers and Elastic Laminae

Elastic fibers are composed of amorphous elastin and fibrous microfibrils (mainly fibrillin-1 and/or -2) [57]. The microfibrils are attached to the cell surface via integrins and the elastin aggregates are integrated along the microfibril scaffold in a process known as elastogenesis [58]. Once crosslinked, mature elastic fibers are insoluble and highly durable, exhibiting a half-life of 74 years [59].

Elastic fibers in the medial layer of arteries are primarily produced by vascular smooth muscle cells [60]. These elastic fibers orientate circumferentially and organize into fenestrated sheets called elastic laminae. The internal elastic lamina defines the boundary between the intima and the media, and the external elastic lamina defines the boundary between the media and the adventitia. Larger arteries have multiple concentric layers of elastic laminae between the internal and external laminae.

## 3. Mechanical Properties of Elastic Laminae

Elastin is highly elastic and imparts unique mechanical properties to elastic laminae within the arterial wall. Elastic laminae will stretch circumferentially when a load is applied and then return to their original configuration when the load is removed. Energy loss is minimal during the loading and unloading cycle, estimated to be 15–20% [61]. The elastic laminae allow for pressure wave propagation in arteries to help the flow of blood. The strain energy stored during systole allows blood to continue flowing downstream during diastole as the elastic arteries recoil. This is especially important for the coronary circulation, which is perfused during diastole.

Atomic force microscopy (AFM) measurements have determined that single elastic fibers have Young’s moduli in the range of 0.3–1.5 MPa [62,63]. Elastin from aortic tissue has a Young’s modulus in the range of 0.1–0.8 MPa and ultimate strain in the range of 100–120% [64,65]. Elastin is several orders of magnitude more compliant than collagen, which has a Young’s modulus in the range of 300–1200 MPa [66,67].

Elastic fibers in the aorta of rabbits are oriented circumferentially (i.e., perpendicular to blood flow) with the exception of the internal elastic lamina, where elastic fibers are oriented longitudinally (i.e., parallel to blood flow) [68]. The circumferentially oriented elastic fibers are able to support the circumferential mechanical stress that arteries experience during systole. The longitudinally oriented elastic fibers are finer and more fenestrated to act as a semi-permeable membrane.

The circumferential mechanical properties of elastin in the descending thoracic aorta of pigs are position-dependent [69]. Elastin is 30% stiffer and 54% stronger near the diaphragm compared to that near the aortic arch. This was explained by a progressive increase in circumferential alignment of elastic fibers along the length of the aorta. The study has also found that the circumferential strain of the aortic wall is relatively constant along the aorta for a given pressure, indicating location-dependent variations in cellular and matrix compositions, the arterial wall and lumen dimensions, and the distribution of arterial wall stress.

The micromechanics of elastic laminae in arteries are determined by reversible structural changes: the folding and unfolding of elastic fibers/elastic laminae at the micro-level and stretching and recoiling of elastin at the nano-level [70]. At an arterial blood pressure level, the elastic fibers and laminae are unfolded, whereas at zero blood pressure (when an arterial specimen is removed), these structures are folded. Likewise, the elastin molecules are elongated at an arterial blood pressure level, whereas these molecules recoil at zero blood pressure. Interestingly, the elastic laminae near the inner portion of the arterial wall are wavier than those in the outer portion of the arterial wall and can therefore unfold to a larger degree when blood pressure increases because of the presence of a more negative stress (a higher compressive stress) in the inner portion. This is a mechanism to accommodate the larger circumferential stretch experienced by the inner portion of the wall in response to an increase in arterial blood pressure. This results in approximately even stretch of the elastic laminae and even stress distribution throughout the arterial wall, avoiding inner-wall stress concentration, which is a condition that potentially causes inner-wall cell injury. These findings were confirmed by a subsequent study using synchrotron-based phase-contrast imaging [71].

A study of biaxial mechanical properties of human arteries demonstrated that most common carotid arteries, subclavian arteries, thoracic aortas, abdominal aortas, and common iliac arteries are stiffer in the longitudinal direction, while most renal arteries are stiffer in the circumferential direction [72]. Elastic fibers were primarily circumferentially oriented and localized to the medial layer in the common carotid artery, subclavian artery, thoracic aorta, and abdominal aorta, reflecting the higher elasticity in the circumferential direction. Elastic fibers were primarily longitudinally oriented and localized to the external elastic lamina in the renal artery, reflecting the higher elasticity in the longitudinal direction. Interestingly, the elastic fibers were also primarily longitudinally oriented and localized to the external elastic lamina in the common iliac artery, opposing the higher elasticity in the circumferential direction. The authors speculated that the longitudinally aligned elastic fibers in the common iliac artery may be necessary to accommodate bending and compression as the hip moves.

A study of human left anterior descending (LAD) coronary arteries found that the intimal and adventitial layers are stiffer in the longitudinal direction compared to the circumferential direction, whereas the reverse is true for the medial layer [73]. These findings are consistent with the predominantly longitudinal orientation of collagen and elastin fibers in the adventitial layer of coronary arteries, whereas these fibers have a more complicated three-dimensional structure without a preferred orientation in the medial layer [74].

## 4. Fundamental Pathogenic Processes in Reconstructed Arteries

### 4.1. Inflammation

Inflammation is a series of processes activated in response to surgical and mechanical injury in reconstructed arteries. Inflammation can be divided into three phases: acute, sub-acute, and chronic phases [1,22,23,24,25,26]. The acute phase starts immediately following an injury and lasts several days. This phase is characterized by the activation of inflammatory mediators, elevation in the endothelial permeability, and leukocyte activation and adhesion. Inflammatory mediators include bradykinin, histamine, and cytokines. Bradykinin is a peptide generated by kallikrein protease-mediated cleavage of plasma kininogen expressed primarily in hepatic cells [75,76]. In blood vessels, bradykinin can cause vascular smooth muscle cell relaxation, resulting in vasodilation and elevation in blood flow to injured areas, and can induce an increase in endothelial permeability, facilitating inflammatory mediator transport across the endothelium and leukocyte adhesion [77]. Bradykinin also causes pain, swelling, and diuresis [78]. Histamine is an amino acid derivative from histidine under the action of histidine decarboxylase in primarily mast cells and basophils [79] and is primarily generated and stored in mast cells and basophils [80]. Upon inflammatory stimulation, histamine can be released to act on vascular endothelial cells to open the tight junction, resulting in an increase in endothelial permeability, a change causing edema. Cytokines are a superfamily of small proteins, expressed and released from primarily leukocytes in response to injury [1,81]. The majority of cytokines, such as interleukin 1α (IL1α), IL2, IL3, IL6, IL12, and chemokines, induce leukocyte activation, adhesion, and extravasation, although several cytokines such as IL10, IL27, and IL35 exert an opposite effect [1]. Overall, during the acute phase, the inflammation-stimulating cytokines are dominant to accelerate inflammatory responses.

The acute inflammatory phase is followed by the sub-acute phase, which lasts for several weeks. This phase is characterized by growth factor upregulation, endothelial cell proliferation and angiogenesis, vascular smooth muscle cell proliferation and migration from the host artery to the reconstructed artery, and over-generation of extracellular matrix components, primarily including the collagen matrix and proteoglycans. Several growth factors, including vascular endothelial growth factors (VEGFs), platelet-derived growth factors (PDGFs), and fibroblast growth factors (FGFs), are commonly expressed and released from vascular cells in response to injury [1]. These growth factors regulate vascular cell proliferation and migration via autocrine and paracrine mechanisms. VEGFs can induce vascular endothelial cell proliferation, an essential process for the repair of lost endothelial cells and angiogenesis. PDGFs and FGFs promote vascular smooth muscle cell proliferation and migration from the host artery to the reconstructed artery [1,82]. In autologous vein-based arterial constructs, smooth muscle cells can also migrate from the venous media to the sub-endothelial space, contributing to neointima formation [32,35]. The growth-factor-activated vascular cells can express and release extracellular matrix components, especially collagen, causing matrix accumulation and fibrosis. The chronic phase of inflammation is characterized by the continuous generation of extracellular matrix from the activated smooth muscle cells and fibroblasts, contributing to the advancement of fibrosis.

### 4.2. Thrombosis

Thrombosis is an acute process initiated in response to endothelial injury, hemorrhage, and exposure to biomaterial- and matrix-based arterial constructs. Thrombosis can start with blood coagulation caused by the formation of insoluble fibrin gels from its soluble precursor fibrinogen. This process requires the formation and action of thrombin, a proteinase that can cleave fibrinogen to generate fibrin. Thrombin arises from its precursor prothrombin, an inactive soluble plasma protein expressed and released from the liver [83], under the action of the proteinase prothrombinase. Injured endothelial cells and fibroblasts can express and release this proteinase [84], thus inducing blood coagulation.

Although blood coagulation is a process necessary to stop hemorrhage in the event of trauma, it causes the formation of thrombi, which are pathological structures composed of fibrin and blood cells, including erythrocytes, leukocytes, and platelets, and are found on the surface of reconstructed arteries [1,85,86]. The fibrin gel established during coagulation can attract and entrap blood cells. The entrapped leukocytes can upregulate and release cytokines that can continuously activate and attract leukocytes from the circulatory system to the fibrin gel [87,88]. This process, together with continuous fibrin gel development and blood cell entrapment, contributes to thrombus development [1,87,88]. An extreme case of endothelial injury in the host artery is endothelial denudation, resulting in the exposure of the supporting extracellular matrix. Platelets can adhere to selected matrix components, facilitating blood coagulation and thrombosis [89,90]. In mild injury, a thrombus grows slowly and can be covered by endothelial cells that are regenerated from surrounding endothelial cells. This endothelialization process prevents fibrin formation, blood cell entrapment, and enlargement of the thrombus. A small thrombus usually does not significantly interfere with blood flow. However, in severe injury, rapid fibrin gel formation and blood cell entrapment can occur, resulting in the formation of massive thrombi that can partially or completely obstruct blood flow and cause acute failure of reconstructed arteries. Furthermore, thrombi are not stable and can detach from the base to form emboli, resulting in blockade of distal arteries and ischemic injury [1]. 

### 4.3. Intimal Hyperplasia

Intimal hyperplasia in reconstructed arteries is cell proliferation to increase the cell density within the intima, primarily involving smooth muscle cells, a process resulting in the formation of neointima [30,91,92]. Neointima is focal in nature, often localized to the junction of the host artery with the reconstructed artery, where anastomosis causes injury, and regions exposed to vortex blood flow, where the level of fluid shear stress is low [32,35,93]. In structure, neointima is composed of leukocytes, platelets, smooth muscle cells, and extracellular matrix (primarily collagen and proteoglycans) with endothelial cells on the surface [1,32,57]. In reconstructed arteries, neointima develops from thrombi, involving smooth muscle cell migration from the host artery (and the vascular wall in the case of vein-based arterial constructs) and endothelialization. The consequence of neointima formation is restenosis and failure of reconstructed arteries.

## 5. Anti-Inflammatory and Anti-Thrombogenic Activities of Elastic Laminae

### 5.1. Elastic Laminae-Based Protection against Arterial Inflammation

The arterial media experience much reduced inflammatory activity compared with the arterial intima in response to a given level of injury. The arterial intima is susceptible to leukocyte infiltration, erythrocyte and platelet deposition, and smooth muscle cell hyperplasia, resulting in inflammation, thrombosis, and intimal hyperplasia, whereas the arterial media are rarely inflicted by these pathological processes. The arterial medial resistance to inflammation, thrombosis, and cell hyperplasia can possibly be attributed to the presence of elastic laminae. Key evidence that supports this concept is the capability of the elastic laminae to suppress leukocyte activities.

The arterial elastic laminae resist leukocyte adhesion. In a cell-culture-based test, the elastic-lamina-rich medial matrix and the collagen-rich adventitial matrix were prepared from the mouse aorta, and the elastic lamina surface was exposed by NaOH treatment [8,9]. The prepared matrix specimens were placed on separate culture dishes. Mouse leukocytes were isolated and cultured on the elastic-lamina-rich and collagen-rich matrix substrates. At selected time points (3, 6, 12, and 24 h), the matrix specimens were removed from the culture dishes and used for counting leukocytes. The density of leukocytes on the surface of the collagen-rich adventitial matrix was about 50 to 105 times higher than that on the elastic-lamina-rich medial matrix from 3 to 24 h of culture. These observations support the concept that the elastic laminae prevent leukocyte adhesion.

The arterial elastic laminae prevent leukocyte migration. In a rat arterial reconstruction model in vivo, allogenic aortic matrix scaffolds were prepared by removing cells and grafted into the aorta [9]. Whereas dense leukocytes were found in the collagen-rich adventitia of the aortic matrix scaffold at 1, 10, and 30 days following aortic grafting, leukocytes were rarely present in the elastic-lamina-rich media of the aortic matrix scaffold. The density of leukocytes within the collagen-rich adventitia was about 2000 to 4000 times higher than that within the elastic-lamina-rich media of the aortic matrix scaffold. An interesting observation was that, even at the end of the aortic matrix scaffold, leukocytes were unable to migrate into the inter-elastic lamina gaps, which were considerably larger than the leukocytes. These observations demonstrate the capability of the elastic laminae to inhibit leukocyte migration.

It is important to note that the arterial elastic laminae and their degradation products, elastin-derived peptides, may behave differently in the regulation of inflammatory responses. Whereas the elastic laminae prevent leukocyte adhesion and infiltration, elastin-derived peptides exert the opposite effect [94]. Selected elastin-derived peptides may induce inflammation-stimulating processes by activating the elastin receptor complex and cathepsin A-neuraminidase 1 complex signaling systems, which contribute to inflammatory responses and atherogenesis [94]. It is possible that the exposure of selected domains of a complete 3D-folded elastin molecule is required for the anti-inflammatory action of the arterial elastic laminae. Selected elastin-derived peptides, on the other hand, may exhibit a pro-inflammatory action when the inhibitory domains are disassembled during elastin degradation.

Another point to address is why the arterial media are more resistant to leukocyte infiltration than the arterial intima. In addition to the difference in extracellular matrix composition as discussed above, different cell types—endothelial cells in the intima and smooth muscle cells in the media—may play distinct roles in the control of inflammatory responses. Injured endothelial cells may facilitate leukocyte adhesion and infiltration, whereas smooth muscle cells may hypothetically inhibit these inflammatory activities. However, the following evidence does not support the anti-inflammatory role of smooth muscle cells. First, when the internal elastic lamina was damaged mechanically, leukocytes were able to migrate into the arterial media in the presence of smooth muscle cells in vivo [9]. Second, in decellularized arterial scaffolds, leukocytes were not able to migrate into the gaps between the elastic laminae, even though the gap width was larger than the leukocyte diameter [9]. Thus, the arterial elastic laminae, but not the smooth muscle cells, resist leukocyte adhesion and infiltration. It should be noted that elastic lamina fragmentation occurs in aged arteries. This change often leads to smooth muscle cell migration from the arterial media to the intima, contributing to neointima formation [95,96]. It remains to be demonstrated whether leukocytes can migrate into the media of aged arteries. 

### 5.2. Elastic Lamina-Mediated Prevention of Vascular Smooth Muscle Cell Proliferation and Neointima Formation

The arterial elastic laminae suppress vascular smooth muscle cell proliferation because of the inhibitory action of elastin [13,15,16]. In mice with elastin gene deficiency (*Eln*−/−), the arterial smooth muscle cells exhibit an over-proliferative phenotype [15]. Humans with elastin gene mutation and elastin deficiency, found in supravalvular aortic stenosis and Williams–Beuren syndrome, express a similar phenotype in large arteries, often associated with excessive smooth muscle cell proliferation and arterial hypertrophy and stenosis, resulting in blood flow obstruction [12,14,15,16,19,21]. In experimental coronary artery restenosis, administration of elastin peptides to the injured artery results in a significant reduction in the rate of neointima formation [13].

Arterial elastic laminae could effectively prevent neointima formation in an arterial matrix implantation model [8]. In this investigation, aortic extracellular matrix scaffolds were harvested from donor rats and prepared to remove cells and expose the basal lamina, internal elastic lamina, or adventitial collagen by NaOH treatment. Matrix scaffolds with the three different surface components were implanted into the aortas of recipient rats and examined at 5, 10, and 20 days. The rate of smooth muscle cell proliferation, evaluated by the BrdU incorporation assay, differed substantially across the three allogenic aortic matrix scaffolds with distinct surface components. The elastic lamina surface exhibited the lowest BrdU index compared with the basal lamina and collagen surfaces at a selected time point. The highest BrdU index was found at the collagen surface. In the same allogenic aortic matrix scaffold implantation model, the elastic lamina surface was associated with the lowest level of neointima compared with the basal lamina and collagen surfaces, whereas the collagen surface was associated with the highest level of neointima. These observations support the concept that the arterial elastic laminae inhibit smooth muscle cell proliferation and neointima formation.

### 5.3. Elastin-Enhanced Actin Filament Generation in Vascular Smooth Muscle Cells

The coexistence of elastic laminae in the arterial media suggests a role for elastin in the development and maintenance of the contractile phenotype of the vascular smooth muscle cells. The building block of elastin, tropoelastin, has been demonstrated to cause the generation of actin filaments in vascular smooth muscle cells [7,97]. One peptide from tropoelastin, VGVAPG, has been suggested as a key element responsible for the myofibrillogenesis of smooth muscle cells [97]. Mechanistically, this short peptide, as well as tropoelastin, can induce actin polymerization by activating the G protein-coupled receptor, i.e., the RhoA signaling pathway [97]. These observations support the concept that the elastic laminae serve as a regulatory factor for the development and maintenance of the contractile phenotype of vascular smooth muscle cells.

The arterial elastic laminae play a role in the development of vascular smooth muscle α actin filaments in bone-marrow-derived CD34-positive cells [10]. Organ- and tissue-specific environmental conditions have long been considered cues that induce stem cell differentiation into specified functional cells. The arterial elastic laminae can create a condition in favor of developing the smooth muscle cell contractile phenotype, characterized by the presence of orderly aligned smooth muscle α actin filaments, which contrasts with the proliferative and synthetic phenotype found in the neointima. In an in vitro test, bone-marrow-derived CD34-positive cells developed smooth muscle α actin filaments when cultured on the surface of the mouse arterial elastic laminae, but these cells did not form smooth muscle α actin filaments when cultured on the adventitial collagen matrix [4]. These observations are consistent with the coexistence of smooth muscle cells with the elastic laminae in the arterial media, but not with the collagen matrix in the adventitia, supporting a role for the elastic laminae in the induction of smooth muscle cells from stem cells as well as in the maintenance of the contractile phenotype of the smooth muscle cells.

### 5.4. Mechanisms of the Inhibitory Action of Elastic Laminae

A fundamental question is how the arterial elastic laminae exert an inhibitory effect on the adhesion, proliferation, and migration of vascular cells and leukocytes. A prior investigation demonstrated that an inhibitory signaling pathway, involving signal regulatory protein α (SIRPα) and Src homology 2 domain-containing protein tyrosine phosphatase-1 (SHP1), potentially mediates the inhibitory action of the arterial elastic laminae [9]. The elastic lamina component, elastin, can bind to and activate SIRPα, a transmembrane receptor that can recruit and activate SHP1, an enzyme capable of dephosphorylating selected substrate proteins, including growth factor receptor protein tyrosine kinases, focal adhesion kinase, and Src homology 2 domain-containing protein tyrosine kinase (Figure 1). Growth factor receptor protein tyrosine kinases transmit growth factor signals to cause cell proliferation and migration, and focal adhesion kinase and Src homology 2 domain-containing protein tyrosine kinase relay matrix-dependent integrin signals to stimulate cell adhesion. Dephosphorylation of protein tyrosine kinases usually suppresses the activity of these kinases as well as the kinase-induced cell activities. Given such actions of the protein tyrosine kinases, the deactivation of these kinases in response to SHP1 results in the inhibition of vascular cell and leukocyte adhesion, proliferation, and migration [9]. In summary, the inhibition of the growth factor protein tyrosine kinase activity by the SHP1 action may promote the development of the contractile phenotype of vascular smooth muscle cells. This concept is consistent with the observation that growth factor-activated proliferative smooth muscle cells in injury-induced neointima (a structure without elastic laminae, but with elevated growth factor signaling actions) exhibit much reduced and more irregularly organized α actin filaments compared with healthy arterial medial smooth muscle cells that reside within the gaps between elastic laminae and are subject to a minimal level of growth factor activity [32,35,36]. However, the molecular regulatory mechanisms downstream to SHP1 need further investigation.

## 6. Application of Elastic Laminae and Elastin-Based Materials to Arterial Reconstruction

Autogenous vein- and arterial graft-based arterial reconstruction is an effective approach for the treatment of occlusive arterial disorders. Autogenous venous and arterial grafts have been considered the most reliable graft types because of their natural properties and performance, although these grafts are prone to thrombosis, inflammation, and intimal hyperplasia, resulting in graft stenosis and failure [98,99,100,101,102,103]. However, due to limited quantity, vascular disease, or prior harvests, suitable autogenous grafting materials are often unavailable. Consequently, researchers have been working intensively over the past several decades to develop reliable tissue-engineered and synthetic-material-based grafts for vascular reconstruction.

A major challenge in the development of effective tissue-engineered and synthetic vascular grafts is the prevention of inflammatory and thrombogenic responses [104,105,106,107]. Numerous synthetic vascular graft types have been developed, but are not suitable for small diameter artery reconstruction due to host inflammatory and thrombogenic responses. Synthetic materials possess poor blood compatibility, often activating leukocytes and causing blood coagulation, which are fundamental processes that lead to inflammation and thrombosis [108].

Tissue-engineering approaches have been used to develop patient-matching cellularized vascular constructs with natural properties [107,109], which are efficacious for replacing failed small diameter arteries. Although few tissue-engineered vascular grafts have attained widespread clinical acceptance, several vascular tissue-engineering strategies have been promoted for developing arterial constructs. In particular, cell-sheet-based approaches have been used to produce natural blood-vessel-like constructs, some of which have been tested in clinical trials with excellent results [108]. For example, long-term results from a recent human study with a cell-sheet-derived graft for hemodialysis access showed great promise as an alternative to synthetic grafts [109,110]. However, these approaches require tedious and complex culture and maturation processing. They also require a patient-matching cell harvest and expansion phase. The need for patient-matching cells extends the preparation time, increases the cost, and shortens the shelf-life, all of which precludes the feasibility of meeting the urgent needs of patients who require off-the-shelf grafts. Furthermore, it may not be possible to reproduce the exquisite material properties inherent to native arterial tissue.

Decellularized arterial matrix scaffolds hold greater promise as implant materials for small-diameter arterial reconstruction [111,112]. These scaffolds possess ideal mechanical properties, porosity, and cell adhesive and regenerative properties. However, native arteries that are decellularized to avoid immunogenic responses no longer have an endothelium, displaying an exposed thin collagen-rich matrix layer (from the basal lamina) to the circulating blood, promoting potent thrombosis and inflammation upon engraftment. Decellularization approaches therefore typically require a patient-matching endothelial layer on the luminal surface of arterial constructs before engraftment [107]. However, endothelial cells grown on arterial constructs are not stable and can detach rapidly when exposed to the arterial blood flow. The field of vascular engineering has not developed an effective arterial construct with an intact native endothelium despite many decades of intensive effort [113]. Furthermore, this approach requires cell harvesting, seeding, and expansion, precluding urgent uses.

To date, materials used for vascular graft engineering, including synthetic polymers and collagen matrices, can cause profound inflammatory and thrombogenic responses when placed in blood contact [104,107,114], contributing to intimal hyperplasia and graft stenosis. Since the arterial elastic-lamina-dominant matrix can effectively prevent leukocyte adhesion and transmigration in matrix-based arterial reconstruction and reduce inflammation and thrombosis [115,116], this matrix may be used to establish a blood contacting layer, as a substitute for an intact endothelium, to alleviate inflammatory and thrombogenic responses in engineered vascular grafts. Several vascular graft-engineering approaches have been developed to exploit the anti-inflammatory and anti-thrombosis properties of elastin [117]. One approach incorporates synthetic elastin or the tropoelastin precursor into engineered materials [118,119,120,121,122,123,124,125]. This approach can produce a synthetic graft with thrombogenic and immunogenic suppressive properties; however, the elastin-containing material lacks the mechanical properties of the native artery that has been optimized through evolution for cell and tissue regeneration/remodeling.

Another approach relies on decellularized arterial elastic laminae. In this approach, the elastic laminae are either completely purified from a donor artery or left interconnected with a portion of the native collagen matrix. In the case of complete elastic lamina purification, a synthetic scaffold is required to support the purified native elastic lamina layer due to the loss of the mechanical strength from the removal of the collagen matrix [126]. In the other case, if the elastic laminae remain interconnected with the native collagen matrix, it is necessary to deplete the basal lamina to expose the internal elastic lamina as the blood-contacting surface while maximizing the preservation of the mural collagen matrix and other native constituents to retain the natural regenerative properties. One approach is to incubate the native artery in an acidic or basic solution for varying times to completely deplete the basal lamina from the blood-contacting surface while reducing the dissolution of mural collagen matrix [8].

All materials developed so far for vascular graft engineering have critical limitations that preclude their widespread clinical acceptance. There is a pressing need to develop clinically feasible, universally applicable, and shelf-stable vascular grafting materials. An ideal vascular graft should match the mechanical characteristics of the native artery. Concurrently, viable vascular grafts must re-establish the properties of a native endothelium at the blood-contacting surface, avoid provoking inflammatory responses, remain shelf-stable over extended time periods prior to use, be easy to handle, and be commercially and clinically feasible. The main challenges have been mimicking a confluent endothelium on the luminal surface of grafting materials and difficulties in generating biocompatible arterial constructs. A future direction is to use decellularized native matrix materials with a uniform elastic lamina blood-contacting surface while retaining the outstanding mechanical properties, porosity, and cell-interaction capability of native arteries. We have made considerable progress in engineering such a vascular grafting material by selective surface collagen depletion in decellularized porcine internal mammary arteries (Figure 2). Once the approach is optimized, the elastin-rich grafts may be used for arterial reconstruction without the need for endothelialization. Acellular grafting materials with a luminal elastic lamina surface are shelf-stable and can be stocked and used to meet the urgent needs of patients.

It is important to address the technical aspect of preparing arterial matrix scaffolds with a luminal elastic lamina surface. We have used a high viscosity acidic gel to restrict matrix depletion activity to the artery luminal surface. The gel (made from polylactic acid or polyglycolic acid) is of sufficient molecular weight to reduce acid diffusion into the arterial wall. Thus, the natural matrix composition and structure of native arteries can be preserved, including mechanical and remodeling properties. Future work along this direction could produce commercially and clinically feasible small-diameter vascular grafts that can be mass-produced at low cost by using donor specimens. These strategies will potentially establish a basis for the development of matrix material-based grafts for small artery reconstruction.

Another potential application for elastin-based materials is to coat arterial stents for preventing in-stent thrombosis and neointima formation. Similar to biomaterial-based arterial constructs, stents can cause thrombosis rapidly following placement, resulting in neointima formation and arterial restenosis [127,128]. This has long been a challenging problem. Coating stents with various agents, such as anticoagulants, biomaterials, antimitotic substances, and corticosteroids [127], has been considered a potentially effective approach for reducing stent-induced neointima formation. However, the current coating agents may cause substantial inflammatory responses (biomaterials) or may only exert anti-thrombogenic and anti-hyperplastic effects for a short period (anticoagulants, antimitotic substances, and corticosteroids). Elastin is an insoluble stable polymer that, if able to firmly adhere to stents, may exert anti-inflammatory, anti-thrombogenic, and anti-hyperplastic effects for a longer time.

## 7. Concluding Remarks

The elastin-based extracellular matrix exhibits an inhibitory effect on leukocyte and vascular smooth muscle cell proliferation, adhesion, and migration. Given that these cell activities contribute to neointima formation in reconstructed arteries, this feature renders the elastin matrix a potential material for constructing the blood-contacting surface of arterial constructs. Preliminary experimental investigations have provided promising results. However, the mechanisms of the inhibitory elastin action remain to be investigated, and it is challenging to coat the luminal surface of an arterial construct with an elastin-based matrix.

## Figures and Tables

**Figure 1 bioengineering-10-00424-f001:**
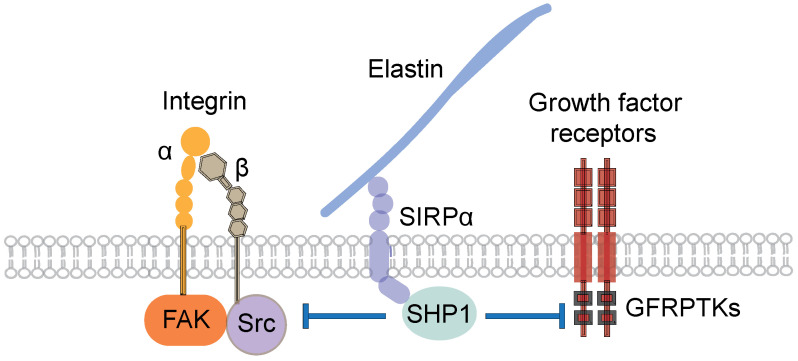
Mechanisms of the inhibitory action of elastin. SIRPα: Signal regulatory protein α. GFRPTK: Growth factor receptor protein tyrosine kinases. FAK: Focal adhesion kinase. Src: Src homology 2 domain-containing protein tyrosine kinase. SHP1: Src homology 2 domain-containing protein tyrosine phosphatase-1.

**Figure 2 bioengineering-10-00424-f002:**
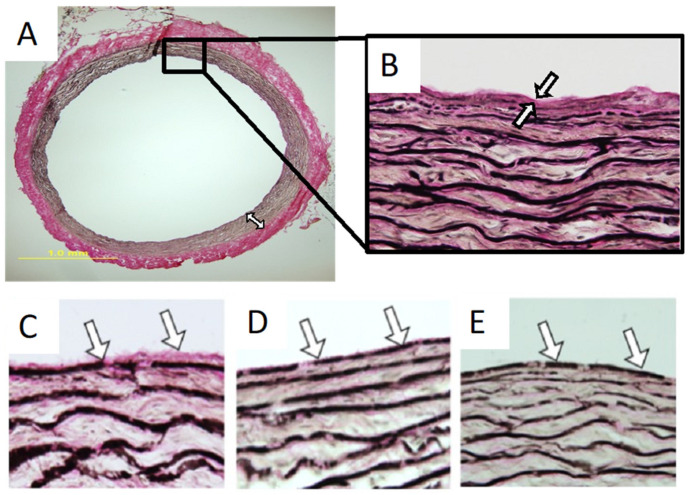
Preparation of arterial matrix scaffolds with a luminal elastic lamina surface. (**A**,**B**) Porcine mammary artery cross section (**A**) and close-up (**B**) showing collagen matrix (red) and abundant elastic laminae (black) in the arterial wall. The thin layer of native luminal surface collagen is identified by arrows. (**C**–**E**) Arterial matrix specimens processed by using high-viscosity acidic gels ((**C**), control gel; (**D**), 2 h treatment in glycolic acid gel; (**E**), 2 h treatment in lactic acid gel). These treatments can selectively remove the internal collagens while preserving the mural collagens. Additional work is required to optimize this approach.

## Data Availability

The data presented in this study are available in Figure 2.

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
