# Peer review of "Anti-Inflammatory and Anti-Thrombogenic Properties of Arterial Elastic Laminae"

_bioengineering, 2023, doi:10.3390/bioengineering10040424_

Round 1

Reviewer 1 Report

This is an interesting review manuscript summarizing the impact of arterial elastin/elastic laminaeon atherogenicity/inflammation/thrombogenicity, and proposing new strategies (including the author strategy)  to design biomaterial-based grafts which would prevent the present limitations of these materials. The manuscript is generally well written and referenced, with interesting proposals, although some elements are missing and some paragraphs are confusing.

General: A comprehensive/synthetic schematic showing the respective actions of elastic laminae and collagen (including the expected actions in potential/future grafted biomaterials) on inflammation, thrombogenicity and atherosclerosis would be very helpful to the readers.

Specific comments:

lines 24-27: " The essential functions of the extracellular matrix are to organize vascular endothelial cells, smooth muscle cells, and fibroblasts into the intima, media, and adventitia of the arterial wall, respectively; provide mechanical strength to the arterial wall ": not only strength: mechanical strength and elasticity.

lines 28-31: " Elastic laminae work with the collagen matrix in an antagonistic manner to control vascular cell and leukocyte adhesion, proliferation, and migration – cell activities directly influencing inflammatory, thrombogenic, and atherogenic processes ": not only that: work in an antagonistic manner to control tissue mechanical properties (elasticity-distensibility regarding elastin, resilience-stiffness regarding collagen), as well as vascular cell and leukocyte adhesion ... .

Lines 31-34: " Whereas the collagen matrix stimulates these cell activities, enhancing inflammatory, thrombogenic, and atherogenic processes, the elastic laminae exert an opposite effect, suppressing these pathogenic processes [2-4]. ": the authors have cited only their own work regarding these effects of collagen, while a substantial number of results has been published by other groups regarding these effects. It would be important to also cite some previous and following works by other teams.

Line 53 "Atherosclerosis can arise directly from a thrombus " and lines 226-227 "Vascular injury can cause the formation of atheromas, which usually arise from thrombi ": often a thrombus could arise from the rupture of an atheroma plaque, and is a major cardiovascular risk regarding atherosclerosis. The claimed reverse process, i.e. atherosclerosis arising from a thrombus, has to be described and referenced more precisely. In lines 226-227, references should be cited to support the description of this reverse process (no reference cited for now). Atheroma often results from long term dyslipidemia and endothelial dysfunction, not thrombi. This has to be described too.

Line 65: elastin is not encoded by the ELN gene (elastin is a polymer): Tropoelastin, the precursor of elastin, is encoded by the ELN gene ... .

Lines 89-90: microfibrils are not only fibrillin-1 and -2: ... fibrous microfibrils (mainly composed of fibrillin-1 and/or -2)

Lines 94-95: " Elastic fibers in the media layer of arteries are primarily produced by vascular  smooth muscle cells [32] ": reference #32, focusing on LDLs and atherosclerosis is probably not the best reference to cite in order to support the role of medial VSMCs in the production of elastic fibers.

Lines 226-233: no reference appears in this long paragraph. Please cite references.

Lines 234-243: This paragraph is puzzling. It seems that the authors confuse the thrombus and the atheroma plaques. The paragraph describes what occurs in the atheroma plaque (within the arterial wall) not in the thrombus (in the lumen). What is indicated in this paragraph is not what is described in the reference to which the paragraph refers (reference #53, which describes what happens in the atheroma plaque, not the thrombus). Could the authors clarify this ?

Lines 270-299: could the authors cite works by other teams related to the anti-inflammatory action of elastin/elastic fibers/elastic laminae ? Would it be possible to cite/discuss works that, on the contrary, indicated that elastin peptides promote inflammation/atherogenesis in the arterial wall (e.g.: Gayral S et al. Elastin-derived peptides potentiate atherosclerosis through the immune Neu1-PI3Kγ pathway. Cardiovasc Res. 2014 Apr 1;102(1):118-27. doi: 10.1093/cvr/cvt336.) ?

Lines 326-340: what is the pertinence and conclusion of experiments performed in bone-marrow cells ? What could these experiments tell about the reaction of vascular smooth muscle cells (VSMCs) to elastic laminae ? These two cell types could react differently to the same stimulus. Wouldn't it be more pertinent to cite experiments performed in VSMCs showing that the presence of more extracellular elastin induces more and more organized smooth muscle alpha-actin ? See : your reference #7 and Karnik et al. Elastin induces myofibrillogenesis via a specific domain, VGVAPG. Matrix Biology 22 (2003) 409–425.

Line 441: wrong format/numbering of the cited reference: [2 Liu et al., 2004]

Author Response

The authors would like to thank the reviewer for evaluating the manuscript and providing helpful comments. The manuscript has been revised based on the review comments. Response to each review comment is presented below. Changes in the manuscript are highlighted in blue.

General comments

Comments: This is an interesting review manuscript summarizing the impact of arterial elastin/elastic laminae on atherogenicity/inflammation/thrombogenicity, and proposing new strategies (including the author strategy) to design biomaterial-based grafts which would prevent the present limitations of these materials. The manuscript is generally well written and referenced, with interesting proposals, although some elements are missing and some paragraphs are confusing. 

      Response: The manuscript has been revised to add missing information, remove confusing concepts, and correct typos.

Comments: A comprehensive/synthetic schematic showing the respective actions of elastic laminae and collagen (including the expected actions in potential/future grafted biomaterials) on inflammation, thrombogenicity and atherosclerosis would be very helpful to the readers.

      Response: A schematic has been added to the manuscript.

Specific comments

Comments: lines 24-27: " The essential functions of the extracellular matrix are to organize vascular endothelial cells, smooth muscle cells, and fibroblasts into the intima, media, and adventitia of the arterial wall, respectively; provide mechanical strength to the arterial wall ": not only strength: mechanical strength and elasticity.

      Response: Thanks. “Elasticity” has been added to this sentence.

Comments: lines 28-31: " Elastic laminae work with the collagen matrix in an antagonistic manner to control vascular cell and leukocyte adhesion, proliferation, and migration – cell activities directly influencing inflammatory, thrombogenic, and atherogenic processes ": not only that: work in an antagonistic manner to control tissue mechanical properties (elasticity-distensibility regarding elastin, resilience-stiffness regarding collagen), as well as vascular cell and leukocyte adhesion ... .

      Response: The authors would prefer not to address more extensively the difference in mechanical properties between the arterial elastic laminae and collagen matrix as this manuscript is focused on the biological aspect of the elastic laminae.

Comments: Lines 31-34: " Whereas the collagen matrix stimulates these cell activities, enhancing inflammatory, thrombogenic, and atherogenic processes, the elastic laminae exert an opposite effect, suppressing these pathogenic processes [2-4]. ": the authors have cited only their own work regarding these effects of collagen, while a substantial number of results has been published by other groups regarding these effects. It would be important to also cite some previous and following works by other teams.

      Response: Several new references have been cited.

Comments: Line 53 "Atherosclerosis can arise directly from a thrombus " and lines 226-227 "Vascular injury can cause the formation of atheromas, which usually arise from thrombi ": often a thrombus could arise from the rupture of an atheroma plaque, and is a major cardiovascular risk regarding atherosclerosis. The claimed reverse process, i.e. atherosclerosis arising from a thrombus, has to be described and referenced more precisely. In lines 226-227, references should be cited to support the description of this reverse process (no reference cited for now). Atheroma often results from long term dyslipidemia and endothelial dysfunction, not thrombi. This has to be described too. 

      Response: Thanks for pointing out this conceptual error. The original manuscript was not well organized with mixing-up of the concepts of atherosclerosis in native arteries and neointima formation in reconstructed arteries. The revised manuscript addresses primarily neointima formation in reconstructed arteries, where thrombi develop first in response to surgical injury, mechanical distension, and exposure of blood to biomaterials, serving as the basis for neointima development. New references have been cited for the concept of neointima formation in reconstructed arteries.

Comments: Line 65: elastin is not encoded by the ELN gene (elastin is a polymer): Tropoelastin, the precursor of elastin, is encoded by the ELN gene ... .

      Response: We have revised this statement to make it clearer that the ELN gene encodes tropoelastin, which is the protein precursor to elastin.

Comments: Lines 89-90: microfibrils are not only fibrillin-1 and -2: ... fibrous microfibrils (mainly composed of fibrillin-1 and/or -2)

      Response: We have revised this statement as suggested by the reviewer.

Comments: Lines 94-95: " Elastic fibers in the media layer of arteries are primarily produced by vascular  smooth muscle cells [32] ": reference #32, focusing on LDLs and atherosclerosis is probably not the best reference to cite in order to support the role of medial VSMCs in the production of elastic fibers.

      Response: We thank the reviewer for identifying this error. We have corrected reference [32] to the originally intended study: Hungerford, J.E., et al., Development of the aortic vessel wall as defined by vascular smooth muscle and extracellular matrix markers. Dev Biol, 1996. 178(2): p. 375-92.

Comments; Lines 226-233: no reference appears in this long paragraph. Please cite references.

      Response: This paragraph describes atherosclerosis in native arteries in the original manuscript. This concept has been removed from the revised manuscript.

Comments: Lines 234-243: This paragraph is puzzling. It seems that the authors confuse the thrombus and the atheroma plaques. The paragraph describes what occurs in the atheroma plaque (within the arterial wall) not in the thrombus (in the lumen). What is indicated in this paragraph is not what is described in the reference to which the paragraph refers (reference #53, which describes what happens in the atheroma plaque, not the thrombus). Could the authors clarify this?

      Response: This paragraph has been removed.

Comments: Lines 270-299: could the authors cite works by other teams related to the anti-inflammatory action of elastin/elastic fibers/elastic laminae ? Would it be possible to cite/discuss works that, on the contrary, indicated that elastin peptides promote inflammation/atherogenesis in the arterial wall (e.g.: Gayral S et al. Elastin-derived peptides potentiate atherosclerosis through the immune Neu1-PI3Kγ pathway. Cardiovasc Res. 2014 Apr 1;102(1):118-27. doi: 10.1093/cvr/cvt336.) ?      

      Response: The different effects of elastic laminae and elastin-derived peptides on inflammatory responses are now discussed in the revised manuscript.

Comments: Lines 326-340: what is the pertinence and conclusion of experiments performed in bone-marrow cells ? What could these experiments tell about the reaction of vascular smooth muscle cells (VSMCs) to elastic laminae ? These two cell types could react differently to the same stimulus. Wouldn't it be more pertinent to cite experiments performed in VSMCs showing that the presence of more extracellular elastin induces more and more organized smooth muscle alpha-actin ? See : your reference #7 and Karnik et al. Elastin induces myofibrillogenesis via a specific domain, VGVAPG. Matrix Biology 22 (2003) 409–425.

      Response: This is a helpful suggestion. The impact of elastin on the development and maintenance of the contractile phenotype of vascular smooth muscle cells is now being discussed in the revised manuscript. We would also like to keep the concept that the arterial elastic laminae stimulate the development of smooth muscle α actin filaments in bone marrow-derived smooth muscle α actin-positive cells. These bone marrow cells may serve as precursors for smooth muscle cells. Their different differentiation behaviors on elastic laminae and collagen matrix may suggest distinct roles for these matrix components in mediating the development of vascular smooth muscle cells.

Comments: Line 441: wrong format/numbering of the cited reference: [2 Liu et al., 2004]

      Response: This has been fixed.

Reviewer 2 Report

The review by Goldman et al. discussed the anti-atherogenic properties of the arterial elastic laminae, suggesting its use as a material for vascular regenerative engineering. The topic is interesting, also considering the wide prevalence of cardiovascular diseases requiring surgical interventions and the use of vascular grafts; however, some points have to be addressed: 

-        In the first paragraph, the authors wrote, "Atherosclerosis can arise directly from a thrombus ….". Thrombosis is a potential complication of the atherosclerotic process, not the "primum movens". The same concept is reported in paragraph 4.3 ("Vascular injury can cause the formation of atheromas, which usually arise from thrombi" or "During thrombosis, low-density lipoproteins (LDLs) can easily access and deposit to thrombi"). According to previous literature, several mechanisms correctly reported by the authors (vascular injury, endothelial dysfunction) might be considered causative factors of the complex multifactorial atherosclerotic process that leads to plaque development and progression. Nevertheless, thrombosis usually represents a complication following plaque rupture. I recommend reconsidering this concept and re-writing these sentences. At least please provide references that confirm these pathological processes, if available. 

-       In paragraph 4.3, the authors reported, "An atheroma can become vulnerable to rupture, a condition potentially resulting in the formation of embolus, a piece of free atheroma that can cause blockage of blood flow in distal arteries – a pathological event known as embolism". Again, according to previous literature, the rupture of the plaque cap might cause thrombosis and not embolism. Then, the thrombus may embolize in a distal arterial segment or provokes the occlusion of the vessel causing ischemia. Please modify the sentence and provide references. 

-      The authors stated that the media layer could be less susceptible to inflammatory degeneration because of the presence of the elastic laminae. This might represent a protective factor; however, the two layers are constituted by different types of cells, endothelial cells in the intima and muscle cells in the media. Could this have a role? Please comment.

-    In arteries, significant changes due to increased inflammation and oxidative stress might develop due to ageing (Mech Ageing Dev 2019; doi: 10.1016/j.mad.2019.111161). Which protective role of the elastic laminae in this setting, thus in the ageing-induced degenerative arterial process?

-       In the third paragraph, the authors reported how the mechanical properties of the elastic laminae might depend on the elastin fibers alignment. They discussed the alignment observed in several arterial segments. Any data regarding the coronary arteries?

-      The book "Liu, S.Q. Cardiovascular Engineering: A Protective Approach. 1st edition. McGraw-Hill, New York, USA, 2020" is the reference from line 39 to line 62, 173 to 180 and 204 to 207. Would it be possible to integrate and cite other articles/reviews about the topic to support what is commented on in these paragraphs? For example: "The acute phase starts immediately following an injury and lasts several days. This phase is characterized by the activation of inflammatory mediators, vasodilation…" can the authors add a valid reference to support it? Again, "This density gradient drives smooth muscle cell migration from the arterial media to the intima, a critical process contributing to atheroma formation and growth", and then "the chronic phase of inflammation is characterized by the continuous generation of extracellular matrix from stimulated smooth muscle…", please add valid references.

-      There are no references to support the sentences from lines 208 to 221, 253 to 268, 271 to 278 and 438 to 480. Please add them.

-        The authors wrote, "In experimental coronary artery restenosis, administration of elastin to the injured artery results in a significant reduction in the rate of neointima formation". Do the authors think that elastin can also play a role in the development of in-stent restenosis and, in particular, in the development of in-stent neo-atherosclerosis (DOI: 10.3390/life12030393)? Please discuss it.

-       The authors discuss engineered vascular grafts. Are there cases or ongoing clinical trials to test their effectiveness? Moreover, are molecules/drugs capable of slowing down atherosclerosis progression by acting on the elastic lamina being studied ever (also in the pre-clinical phase)? Please discuss these points.

-       Please consider adding a figure summarizing all the favourable and anti-atherogenic properties of the elastic laminae to improve the review's readability. 

-       The English language can be improved throughout the manuscript, and several typos should be corrected.

Author Response

The authors would like to thank the reviewer for evaluating the manuscript and providing helpful comments. The manuscript has been revised based on the review comments. Response to each review comment is presented below. Changes in the manuscript are highlighted in blue.

Comments; The review by Goldman et al. discussed the anti-atherogenic properties of the arterial elastic laminae, suggesting its use as a material for vascular regenerative engineering. The topic is interesting, also considering the wide prevalence of cardiovascular diseases requiring surgical interventions and the use of vascular grafts; however, some points have to be addressed: 

In the first paragraph, the authors wrote, "Atherosclerosis can arise directly from a thrombus ….". Thrombosis is a potential complication of the atherosclerotic process, not the "primum movens". The same concept is reported in paragraph 4.3 ("Vascular injury can cause the formation of atheromas, which usually arise from thrombi" or "During thrombosis, low-density lipoproteins (LDLs) can easily access and deposit to thrombi"). According to previous literature, several mechanisms correctly reported by the authors (vascular injury, endothelial dysfunction) might be considered causative factors of the complex multifactorial atherosclerotic process that leads to plaque development and progression. Nevertheless, thrombosis usually represents a complication following plaque rupture. I recommend reconsidering this concept and re-writing these sentences. At least please provide references that confirm these pathological processes, if available. 

      Response: Thanks for pointing out this conceptual error. The original manuscript was not well organized with mixing-up of the concepts of atherosclerosis in native arteries and neointima formation in reconstructed arteries. The revised manuscript addresses primarily neointima formation in reconstructed arteries, where thrombi develop first in response to surgical injury, mechanical distension, and exposure of blood to biomaterials, serving as the basis for neointima development. New references have been cited for the concept of neointima formation in reconstructed arteries.

Comments: In paragraph 4.3, the authors reported, "An atheroma can become vulnerable to rupture, a condition potentially resulting in the formation of embolus, a piece of free atheroma that can cause blockage of blood flow in distal arteries – a pathological event known as embolism". Again, according to previous literature, the rupture of the plaque cap might cause thrombosis and not embolism. Then, the thrombus may embolize in a distal arterial segment or provokes the occlusion of the vessel causing ischemia. Please modify the sentence and provide references. 

      Response: The revised paper now addresses neointima formation in reconstructed arteries. The concept of atherosclerosis in native arteries has been removed. New references have been cited.

Comments: The authors stated that the media layer could be less susceptible to inflammatory degeneration because of the presence of the elastic laminae. This might represent a protective factor; however, the two layers are constituted by different types of cells, endothelial cells in the intima and muscle cells in the media. Could this have a role? Please comment.

      Response: It is possible that different vascular cell types may contribute to the susceptibility to inflammatory responses. However, two lines of evidence may not support an anti-inflammatory role for smooth muscle cells. (1) When the internal elastic lamina was damaged mechanically, leukocytes were able to migrate into the arterial media in the presence of smooth muscle cells in vivo; and (2) In decellularized arterial scaffolds, leukocytes were not able to migrate into the gaps between the elastic laminae, even though the gap width was larger than the leukocyte diameter. These observations support the concept that the elastic laminae, but not the smooth muscle cells, can exert an anti-inflammatory effect. This is now discussed in the revised manuscript.

Comments: In arteries, significant changes due to increased inflammation and oxidative stress might develop due to ageing (Mech Ageing Dev 2019; doi: 10.1016/j.mad.2019.111161). Which protective role of the elastic laminae in this setting, thus in the ageing-induced degenerative arterial process?

      Response: The authors would prefer not to discuss this topic as it was not directly pertinent to arterial reconstruction and the authors are not familiar with this topic.

Comments: In the third paragraph, the authors reported how the mechanical properties of the elastic laminae might depend on the elastin fibers alignment. They discussed the alignment observed in several arterial segments. Any data regarding the coronary arteries?

      Response: We thank the reviewer for this insightful question.  We have added a discussion of a study of the mechanical properties of each layer of human coronary arteries [Holzapfel, et al., PMID 16006541] and related this to fiber alignment [Chen et al., PMID 27086118].

Comments: The book "Liu, S.Q. Cardiovascular Engineering: A Protective Approach. 1st edition. McGraw-Hill, New York, USA, 2020" is the reference from line 39 to line 62, 173 to 180 and 204 to 207. Would it be possible to integrate and cite other articles/reviews about the topic to support what is commented on in these paragraphs? For example: "The acute phase starts immediately following an injury and lasts several days. This phase is characterized by the activation of inflammatory mediators, vasodilation…" can the authors add a valid reference to support it? Again, "This density gradient drives smooth muscle cell migration from the arterial media to the intima, a critical process contributing to atheroma formation and growth", and then "the chronic phase of inflammation is characterized by the continuous generation of extracellular matrix from stimulated smooth muscle…", please add valid references.

      Response: New references have been cited. Please note that the revised manuscript is now focused on neointima formation in reconstructed arteries instead of atherosclerosis in native arteries.

Comments: There are no references to support the sentences from lines 208 to 221, 253 to 268, 271 to 278 and 438 to 480. Please add them.

      Response: New references have been cited for the section 4.2. Thrombosis (lines 208 to 221). The sections from lines 253 to 268 and 271 to 278 have been removed. Sentences from 438 to 454 are general comments that reflect the opinions and experience of the authors. Sentences from 454 to 480 are based on new data shown in Figure 2 and not published elsewhere.

Comments: The authors wrote, "In experimental coronary artery restenosis, administration of elastin to the injured artery results in a significant reduction in the rate of neointima formation". Do the authors think that elastin can also play a role in the development of in-stent restenosis and, in particular, in the development of in-stent neo-atherosclerosis (DOI: 10.3390/life12030393)? Please discuss it.

      Response: The authors think that elastin may play a protective role against in-stent neo-atherosclerosis. This concept is now discussed in the revised manuscript.

Comments: The authors discuss engineered vascular grafts. Are there cases or ongoing clinical trials to test their effectiveness? Moreover, are molecules/drugs capable of slowing down atherosclerosis progression by acting on the elastic lamina being studied ever (also in the pre-clinical phase)? Please discuss these points.

      Response:  We are not aware of any clinical trials of arterial reconstruction involving elastic laminae or elastin-based materials and any drugs under development to reduce atherosclerosis by acting on the elastic lamina.

Comments: Please consider adding a figure summarizing all the favourable and anti-atherogenic properties of the elastic laminae to improve the review's readability. 

      Response: A graphical abstract has been presented in the revised manuscript.

Comments: The English language can be improved throughout the manuscript, and several typos should be corrected.

      Response: The manuscript has been revised to correct typos and grammatical errors.

Reviewer 3 Report

I think it is a very good paper and for me Editors should consider for acceptance after minor revisions (i.e. minor spelling checks).

Author Response

The authors would like to thank the reviewer for evaluating the manuscript and providing helpful comments. The manuscript has been revised based on the review comments. Response to each review comment is presented below. Changes in the manuscript are highlighted in blue.

Comments: I think it is a very good paper and for me Editors should consider for acceptance after minor revisions (i.e. minor spelling checks).

      Response: Thanks for the comments. The manuscript has been revised to correct typos and grammatical errors.

Reviewer 4 Report

The authors have provided an excellent and comprehensive review regarding the anti-atherogenic properties of arterial elastic laminae.

The information provided has been adequately selected and substantiated with 80 state-of-the-art references. These have been properly discussed in regard to already obtained results of other scientists, and put together in well-constructed and logical sections.

The clinical relevance has been clearly underlined given that the elastin-based extracellular matrix exhibits an inhibitory effect on leukocyte and vascular smooth muscle cell proliferation, adhesion, and migration. Since these cell activities contribute to neointima formation in reconstructed arteries, the authors suggest that this feature renders elastin matrix a potential material for constructing the blood-contacting surface of vascular grafts. Although further investigations are required, the current knowledge provides a good direction for successful use of elastin-based extracellular matrix in the future.  

Author Response

The authors would like to thank the reviewer for evaluating the manuscript and providing helpful comments. Please note that changes in the manuscript are highlighted in blue.

Comments: The authors have provided an excellent and comprehensive review regarding the anti-atherogenic properties of arterial elastic laminae.

The information provided has been adequately selected and substantiated with 80 state-of-the-art references. These have been properly discussed in regard to already obtained results of other scientists, and put together in well-constructed and logical sections.

The clinical relevance has been clearly underlined given that the elastin-based extracellular matrix exhibits an inhibitory effect on leukocyte and vascular smooth muscle cell proliferation, adhesion, and migration. Since these cell activities contribute to neointima formation in reconstructed arteries, the authors suggest that this feature renders elastin matrix a potential material for constructing the blood-contacting surface of vascular grafts. Although further investigations are required, the current knowledge provides a good direction for successful use of elastin-based extracellular matrix in the future.  

Response: Thanks for the comments.

Round 2

Reviewer 1 Report

The authors have satisfactorily  responded to te reviewer comments.

Author Response

Reviewer 1

Comments: The authors have satisfactorily responded to te reviewer comments.

            Response: Thanks for reviewing the manuscript.

Reviewer 2 Report

The new version of the manuscript focuses on the potential anti-inflammatory and anti-thrombogenic properties of the arterial elastic laminae resulting in the prevention of neointima formation in reconstructed arteries. The role of elastic laminae in the development of native atherosclerotic plaques has been removed. 

In this revised version, the authors addressed most of the concerns.
However, the graphical abstract should be improved; it should summarize all favourable properties of the elastic laminae in a schematic drawing, which must be clear and immediate for the reader.

Author Response

Reviewer 2

Comments: The new version of the manuscript focuses on the potential anti-inflammatory and anti-thrombogenic properties of the arterial elastic laminae resulting in the prevention of neointima formation in reconstructed arteries. The role of elastic laminae in the development of native atherosclerotic plaques has been removed.

In this revised version, the authors addressed most of the concerns.

However, the graphical abstract should be improved; it should summarize all favourable properties of the elastic laminae in a schematic drawing, which must be clear and immediate for the reader.

         Response: Thanks for reviewing the manuscript and providing insightful comments. The graphic abstract has been revised.